# Sustained Myocarditis following Messenger RNA Vaccination against Coronavirus Disease 2019: Relation to Neutralizing Antibody and Amelioration by Low-Dose Booster Vaccination

**DOI:** 10.3390/jcm12041421

**Published:** 2023-02-10

**Authors:** Koji Miyazaki, Toshiharu Fujii, Kikue Mori, Ryuichi Tamimoto, Hirofumi Nagamatsu, Tsutomu Murakami, Yasunori Cho, Shinya Goto, Hidezo Mori

**Affiliations:** 1Department of General Internal Medicine, Tokai University Hachioji Hospital, 1838 Ishikawacho, Hachioji, Tokyo 192-0032, Japan; 2Shoju Sagamihara Clinic, 820-1-101 Harataima, Sagamihara 252-0336, Japan; 3Department of Cardiovascular Medicine, Tokai University Hachioji Hospital, 1838 Ishikawacho, Hachioji, Tokyo 192-0032, Japan; 4Department of Nutritional Science, Faculty of Applied Bioscience, Tokyo University of Agriculture, 1-1-1 Sakuragaoka, Setagaya-ku, Tokyo 156-8502, Japan; 5Mori Clinic, 541-12 Wasedatsurumakicho Shinjuku-ku, Tokyo 259-1143, Japan; 6Department of Cardiovascular Surgery, Tokai University School of Medicine, 143 Simokasuya, Isehara 259-1193, Japan; 7Department of Cardiovascular Medicine, Tokai University School of Medicine, 143 Simokasuya, Isehara 259-1193, Japan

**Keywords:** coronavirus disease 2019 (COVID-19), vaccine, myocarditis, heart failure, neutralizing antibody

## Abstract

We recently reported that sub-acute myocarditis occurred following the initial two doses of messenger RNA-based vaccination against coronavirus disease 2019 (0.3 mL Comirnaty^®^) in elderly Japanese patients with cardiac dysfunction. The present retrospective study of 76 patients revealed that myocarditis following the initial doses persisted for 12 months, was associated with low levels of neutralizing antibodies, and was ameliorated by reducing the third vaccine dose. Low neutralizing antibody levels (<220 U/mL) after the initial doses were an independent predictor of persistent clinical events, defined as death or marked changes in brain natriuretic peptide levels. When the third dose was reduced (0.1 mL), changes in brain natriuretic peptide levels were significantly smaller (*p* = 0.02, *n* = 25), no deaths occurred due to heart failure, and neutralizing antibody levels increased 41-fold (*p* < 0.001) compared with the initial doses. Reduced booster doses could facilitate the worldwide distribution of messenger RNA vaccines.

## 1. Introduction

Messenger RNA (mRNA)-based vaccination against coronavirus disease 2019 (COVID-19) prevents the spread of infection and reduces disease severity [1,2,3]. Because the mortality of patients with heart failure who are hospitalized with COVID-19 is very high [4], vaccination is strongly recommended for these patients [5]. Myocarditis is one of the major adverse effects of COVID-19 mRNA vaccination. COVID-19 mRNA vaccination-induced myocarditis has been reported to occur 2–3 days after vaccination and to be rare and mild in severity with complete recovery within a short period [6,7,8,9,10,11,12,13]. However, we reported the occurrence of severe and sustained cardiac adverse events, possibly due to autoimmune myocarditis, in the subacute phase (1–6 months) following administration of the first and second vaccination doses in elderly patients with an mRNA-based vaccine against COVID-19 (0.3 mL Comirnaty^®^: BioNTech, Mainz, Germany and Pfizer, New York, USA) [14].

In the present study, we aimed to determine whether this possible autoimmune myocarditis is reversible, whether it is related to neutralizing antibody levels against COVID-19, and whether it can be ameliorated by lowering the dose of booster vaccinations (0.1 mL Comirnaty^®^) by retrospectively analyzing the clinical course of 76 elderly Japanese patients with cardiac disorders for 12 months after the administration of the first two doses.

## 2. Methods

### 2.1. Patient Selection

This retrospective observational study involved 76 elderly Japanese patients (median age: 83.5 years; interquartile range: 79.8–88.5 years) with cardiovascular disorders (Table 1). They all received the first dose of an mRNA-based vaccine against COVID-19 (0.3 mL Comirnaty^®^; BioNTech and Pfizer) between 23 April and 30 July 2021, followed by a second dose of the same vaccine 3–4 weeks later. We measured their serum anti-COVID-19 neutralizing antibody levels at 157 ± 42 days after the first vaccination before they received a third dose. All patients included in this study were treated by one of the authors (H. Mori) at home or care facilities (74 patients) or at the outpatient clinic (2 patients) of Tokai University Hachioji Hospital (Hachioji, Japan) during the observation period (from the day of the first vaccination to the date of death or 30 June 2022). The study protocol was approved by the ethics committee of Tokai University School of Medicine (8 July 2022, Approval Number 22R-043).

### 2.2. Definitions

The observation period was divided into three periods: Period 1, days 1–125 after the first vaccination; Period 2, day 126 until the day before the third vaccination (258 ± 24 days after the first vaccination in 52 patients with the third vaccination and an equivalent period for 24 patients who did not receive a third vaccination, including 16 patients for whom the third vaccination was withheld); and Period 3, from the end of Period 2 until 30 June 2022. To normalize brain natriuretic peptide (BNP) and N-terminal pro-brain natriuretic peptide (NT-pro BNP) data, these values were divided by their upper normal limit (18.4 pg/mL and 125 pg/mL, respectively) to give the BNP ratio. In addition, the BNP ratio prior to the first vaccination was defined as the pre-BNP ratio [14]. The maximum values of Periods 1, 2, and 3 were used to calculate the BNP ratio in each period. A significant clinical event was defined as death from any cause or worsening heart failure defined as an increase in the BNP ratio from the baseline of more than 10-fold over the upper limit of normal (hereafter, “change in the BNP ratio > 10”). 

### 2.3. Study Protocol

To determine whether vaccination-induced clinical events are reversible, we compared changes in clinical events and laboratory data before and after vaccination for 12 months. Thereafter, multivariate analysis was performed to determine the risks linked to the vaccination of elderly Japanese patients with cardiac disorders. Clinical data analysis included BNP or NT-pro BNP, serum creatinine, blood urea nitrogen, aspartate aminotransferase, alanine aminotransferase, hemoglobin A1c, C-reactive protein, troponin T, and anti-severe acute respiratory syndrome coronavirus 2 (SARS-CoV-2)-neutralizing antibodies (Table 1). Anti-SARS-CoV-2 neutralizing antibodies were measured using an Elecsys Anti-SARS-CoV-2 S assay (Roche Diagnostics, Rotkreuz, Switzerland) [15].

We administered the third vaccination at the normal dose (0.3 mL Comirnaty^®^) to 27 low-risk patients (a median BNP ratio in Period 2 of 1.58). To avoid worsening of heart failure after the third vaccination, we administered it at a reduced dose (0.1 mL Comirnaty^®^) to 25 medium-risk patients (a median BNP ratio in Period 2 of 9.0) or withheld the vaccination in 16 high-risk patients (a median BNP ratio in Period 2 of 26.6) based on our previous report [14]. We explained the findings of our previous study to all 68 patients and they provided written or oral informed consent to receive the modified vaccination regimen. Five patients died and three patients dropped out before the third vaccination.

To evaluate the safety and effectiveness of low-dose third vaccination in the 25 medium-risk patients, the changes in the BNP ratio in Period 3 from Periods 1 and 2 (Periods 1–2) and neutralizing antibody levels in Period 3 (after the third vaccination) were compared with the changes in the BNP ratio in Periods 1–2 from baseline and neutralizing antibody levels in Period 1 or 2.

### 2.4. Statistical Analysis

Continuous variables are shown as the median (interquartile range) or mean ± standard deviation. The Wilcoxon rank-sum test or unpaired *t*-test was used to compare unpaired continuous variables. For the comparison of paired continuous variables, the Wilcoxon signed-rank test, paired *t*-test, or Friedman test was used. Fisher’s exact test was used to analyze differences in categorical variables. The time to event was analyzed using a Cox proportional-hazards model. Clinical events were defined as a composite of change in the BNP ratio > 10 from baseline or death from any cause. The proportional-hazards assumption was tested using Schoenfeld residuals after fitting a model with the Cox proportional-hazards model. Hazard ratios with 95% confidence intervals were described as crude and adjusted hazard ratios for possible confounders, namely, the pre-BNP ratio, age, sex, body mass index, serum creatinine, and anticoagulation therapy.

## 3. Results

A high incidence of clinical events was observed throughout Periods 1, 2, and 3 (approximately every 4 months after the first and second vaccinations, Table 1). The median BNP ratio significantly increased from 4.5 at baseline (pre-BNP ratio) to 9.8 in Period 1 after the first and second vaccinations (*p* < 0.001, Table 1) In Periods 2 and 3, the BNP ratio remained high at 6.4 and 6.1, respectively. There were no significant changes in the BNP ratio among the three periods. Liver and renal functions did not change significantly throughout the observation period. Troponin T and C-reactive protein levels were significantly increased, but minimal, in Period 3 (Table 1). There were no deaths in Period 1, five in Period 2, and five in Period 3. In all five patients who died in Period 2 and three of the five patients who died in Period 3, the cause of death was a worsening of congestive heart failure associated with severe contractile deterioration, as demonstrated by echocardiography, but without evidence of myocardial ischemia and/or infarction such as serum troponin T elevation and abnormal Q waves or ST-T changes in electrocardiography. In addition, these deaths were frequently (6/8 cases) associated with an elevation of C-reactive protein levels (2.63–36.0 mg/dL) without leukocytosis, including two patients with non-bacterial pneumonia. In Period 3, the other two patients died of brainstem infarction and acute myocardial infarction, respectively, without worsening of congestive heart failure. Changes in the BNPBNP ratio > 10 compared with baseline were noted in 9 of 76 cases (12%), 13 of 76 cases (17%), and 8 of 68 cases (12%) in Periods 1, 2, and 3, respectively.

In addition to a high BNP ratio at baseline (>4.0), low neutralizing antibody levels against COVID-19 (<220 U/mL) were a significant risk factor for clinical events (Figure 1). In patients with low neutralizing antibody levels (<220 U/mL) and decreased cardiac function before the first vaccination indicated by an elevated BNP ratio at baseline (>4.0), the number of clinical events continued to increase throughout Periods 1 to 3 (Figure 1, log-rank test). The time course of clinical events was not affected by the third vaccination, which was performed 258 ± 24 days after the first vaccination (shown above the Kaplan–Meier curves in Figure 1). Multivariate analysis confirmed that decreased cardiac function (BNP ratio at baseline > 4.0) before the first and second vaccinations, low neutralizing antibody levels (<220 U/mL) after the first and second vaccinations, and anticoagulation therapy were independent predictors of clinical events (Table 2). The presence of low neutralizing antibody levels (<220 U/mL) was associated with changes in the BNP ratio > 10 in Periods 1–2 compared with baseline. In 17 patients with changes in the BNP ratio > 10 in Periods 1–2 compared with baseline, 14 patients had neutralizing antibody levels < 220 U/mL (82%, *p* = 0.03, Table 3A). There was no significant association between the presence of neutralizing antibody levels < 220 U/mL and a BNP ratio at baseline > 4.0 (*p* = 0.35, Table 3B) or a BNP ratio > 10 in Periods 1–2 (*p* = 0.17, Table 3C). 

In the 25 medium-risk patients who received the normal-dose first and second vaccinations and low-dose third vaccination, the change in the BNP ratio after the third vaccination (−1.54 ± 5.77) was significantly decreased (*p* = 0.02) compared with that after the first and second vaccinations (4.98 ± 7.35) (Figure 2A,B) according to baseline control analysis. This suggests that the low-dose third vaccination ameliorated the risk of cardiac dysfunction compared with the normal-dose first and second vaccinations. In the 27 low-risk patients who received the normal-dose third vaccination, the change in BNP ratio (0.37 ± 2.2) was not significantly different (*p* = 0.99) compared with that after the first two doses (0.37 ± 1.2).

The median (interquartile range) of neutralizing antibody levels in the 25 low-dose third-vaccination patients and the 27 normal-dose third-vaccination patients were 186 (53.4–655) and 250 (137–516) after the first and second vaccinations and 7670 (3900–17,400) and 8140 (4590–13,600) after the third vaccination, respectively.

After the third vaccination, median neutralizing antibody levels increased by approximately 41- and 33-fold (*p* < 0.001, *p* < 0.001, respectively) compared with after the first and second vaccinations in the low-dose patients and normal-dose patients, respectively, but their levels were not significantly different between the two groups after the first and second vaccinations (*p* = 0.42) and after the third vaccination (*p* = 0.88). Among the 25 medium-risk patients who received the low-dose third vaccination, 1 patient died of brainstem infarction and 1 died of acute myocardial infarction in Period 3, but no patients died from worsening heart failure. In the 16 high-risk patients who did not receive a third vaccination, the event ratio was 25% (4 cases of changes in the BNP ratio > 10 compared with baseline, but no deaths) in Period 2. In Period 3, three patients died of severe congestive heart failure that persisted from Period 2, but no patients newly developed an increase in the BNP ratio > 10 compared with baseline.

## 4. Discussion

The results of this study suggest that possible autoimmune myocarditis following COVID-19 mRNA vaccination persists for up to 12 months and that neutralizing antibody levels < 220 U/mL after the first two doses is another risk factor for clinical events. We also found that a low-dose third vaccination ameliorates the risk of cardiac dysfunction compared with normal-dose vaccination, but without losing its effectiveness in preventing infection.

A high incidence of cardiac events defined as death and/or changes in the BNP ratio > 10 compared with baseline persisted throughout Periods 1 to 3, approximately 12 months after the administration of the first two doses (Table 1). Five patients died in Period 1 after the first and second vaccinations, as reported previously [6]; however, they were not included in the present study because of the lack of neutralizing antibody measurement. Therefore, the death rate was almost constant in Periods 1 to 3 after the administration of the first two doses. In addition to a high BNP ratio at baseline, low neutralizing antibody levels (<220 U/mL) against COVID-19 were a significant risk factor for clinical events (Figure 1 and Table 2). As suggested by Murphy and Longo [16,17], autoimmune myocarditis, specifically anti-idiotype antibody-dependent myocarditis, is the most likely explanation for this persistent myocardial involvement without an increase in serum troponin T levels or electrocardiographic changes.

The frequent occurrence of elevated C-reactive protein levels in the patients who died is also compatible with this hypothesis. Furthermore, low neutralizing antibody levels were associated with the change in the BNP ratio in Periods 1–2 compared with baseline (deterioration of cardiac function after the first two doses), but not with the BNP ratio at baseline (severity of cardiac dysfunction before the first two doses). Therefore, the low levels of neutralizing antibodies would not be due to the poor production of antibodies associated with severe heart failure but would be related to vaccination-induced autoimmune myocarditis and the level of immune complex formation. In other words, the formation of immune complexes of neutralizing antibodies with their anti-idiotype antibodies would reduce the measurable levels of neutralizing antibodies. The clinical presentation of cardiac dysfunction in our study was completely different from that of the acute myocarditis observed after COVID-19 vaccination mainly in young people [6,7,8,9,10,11,12,13]. The incidence and severity of myocarditis after mRNA-based vaccination against COVID-19 are reported to be rare and mild, which is not consistent with the findings in our study. Vaccination-induced myocarditis occurs within 2–3 days after vaccination and is accompanied by chest pain, elevation of serum troponin level, and ST-segment and T-wave changes on the electrocardiogram. In most cases, the severity is mild and complete recovery is made. On the other hand, the myocardial dysfunction reported in our study occurred more than 30 days after vaccination without any of the aforementioned clinical signs of myocarditis. The severity of the disease was often severe with high mortality and was sustained for up to 12 months. The mechanisms of COVID-19 mRNA vaccination-induced myocarditis is hypothesized to be due to autoimmunity triggered by molecular mimicry and/or overstimulation of the innate immune system by RNA molecules in the vaccine [12,13]. As described above, we proposed anti-idiotype antibodies-dependent myocarditis as a cause of this sustained and silent myocardial dysfunction in our previous paper. The predictive value of low neutralizing antibody levels for events and its association with the extent of cardiac deterioration shown in this paper would be additional evidence for the involvement of anti-idiotype antibodies-dependent myocarditis in the pathogenesis of this sustained and silent myocardial dysfunction.

In the 25 patients who received the normal-dose first and second vaccinations and low-dose third vaccination, the extent of the increase in neutralizing antibody levels was profoundly augmented following the third vaccination, while the changes in the BNP ratio (Figure 2) decreased compared with that after the first and second vaccinations. However, the death of two patients and extremely high neutralizing antibody levels after the low-dose third vaccination suggested that it may be beneficial to lower the dose further. As this was an observational study, a large-scale double-blind randomized study is needed to confirm our hypothesis and to devise safer and more effective ways to provide booster vaccinations.

One may argue that death from the natural course of diseases in these patients could not be excluded from the present study. In the previous 4 years, the cardiac death rate every 6 months among patients treated by the co-author at home or in care facilities was 5.0–7.1% (seven terms). The number of cardiac deaths in the previous 6 months (May 1 to October 31, 2021) was 12 patients (13.5%), including 6 patients who died after vaccination with an mRNA-based vaccine against COVID-19. The number of cardiac deaths among patients treated at home or in care facilities who did not receive the vaccine during the same period was 6 patients (6.7%). In the present study, the death rate over 8 months (periods 2 and 3) was 10/76 (13%).

In conclusion, the COVID-19 mRNA vaccination-induced cardiac events may be due to autoimmune myocarditis that persisted over a period of 12 months after the first vaccination and related to low neutralizing antibody levels against COVID-19. A low-dose (one-third) third vaccination ameliorated the risk of cardiac deterioration without losing its effectiveness in preventing viral infection. Low-dose vaccination is also preferable for the worldwide distribution of mRNA vaccines.

## Figures and Tables

**Figure 1 jcm-12-01421-f001:**
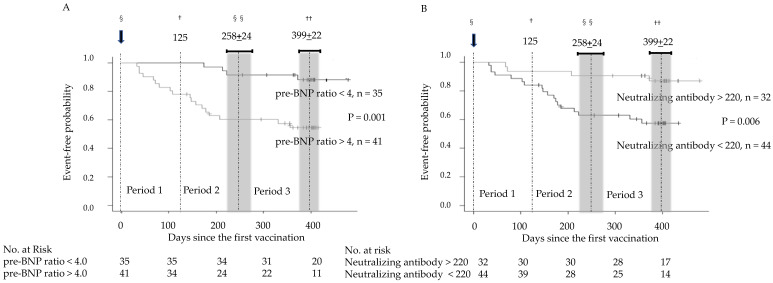
Event-free probability during the clinical course according to brain natriuretic peptide ratio at baseline (pre-BNP) and neutralizing antibody levels. (**A**) A high brain natriuretic peptide ratio at baseline > 4.0 and (**B**) low neutralizing antibody levels < 220 U/mL were significant predictors of clinical events. The number of clinical events continued to increase throughout Periods 1 to 3, and their time course was not affected by the third vaccination. §, first vaccination at Day 0; ^†^, border between Periods 1 and 2 at Day 125 from ^§^; ^§§^, timing of the third vaccination, which is the border between Periods 2 and 3 at 258 ± 24 days from ^§^; ^††^, end of Period 3 at 399 ± 22 days from ^§^.

**Figure 2 jcm-12-01421-f002:**
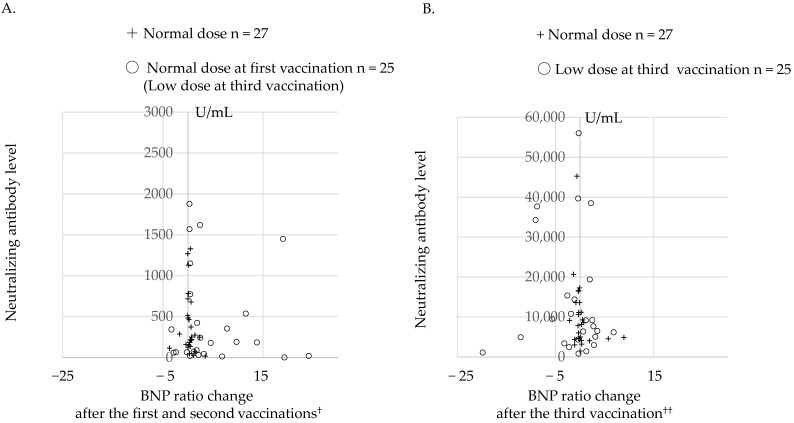
Relationships between neutralizing antibody levels and brain natriuretic peptide (BNP) ratio changes (**A**) after the first and second vaccinations and (**B**) after the third vaccination. +, 27 patients who received the normal dose of the first, second, and third vaccinations; ○, 25 patients who received the normal dose of the first and second vaccinations and low dose of the third vaccination; ^†^, BNP ratio in Periods 1–2 minus the BNP ratio at baseline; ^††^, BNP ratio in Period 3 minus the BNP ratio in Periods 1–2.

**Table 1 jcm-12-01421-t001:** Baseline characteristics and time course of the clinical data.

Characteristic	Baseline	Period 1 ^†^	Period 2 ^††^	Period 3 ^†††^
Age, years (range)	83.5 (79.8–88.5)			
Males, *n* (%)	32 (42.1%)			
Height, cm (range)	155.0 (145.0–162.3)			
Weight, kg (range)	48.5 (42–59)			
Body mass index (range)	21.1 (18.6–23.3)			
Hypertension, *n* (%)	43 (57%)			
Dyslipidemia, *n* (%)	41 (54%)			
Diabetes mellitus, *n* (%)	25 (33%)			
Old myocardial infarction, *n* (%)	11 (14%)			
Valvular heart disease, *n* (%)	24 (32%)			
Hypertensive heart disease, *n* (%)	56 (74%)			
Anticoagulation, *n* (%)	26 (34%)			
Warfarin, *n* (%)	14 (18%)			
Direct oral anticoagulants, *n* (%)	12 (16%)			
Serum creatinine, mg/dL (range), missing *n*	0.9 (0.7–1.1), 0	0.9 (0.7–1.1), 12 ^NS^	0.9 (0.7–1.1), 5 ^NS^	0.9 (0.7–1.1), 6 ^NS^
Urea nitrogen, mg/mL (range), missing *n*	19 (16–24), 2	21 (16–25), 13 ^NS^	22 (17–27), 5 ^NS^	20 (17–26), 6 ^NS^
Aspartate aminotransferase, IU/L (range), missing *n*	21 (17–26), 4	22 (17–27), 14 ^NS^	22 (18–26), 7 ^NS^	21 (17–26), 7 ^NS^
Alanine aminotransferase, IU/L (range), missing *n*	14 (11–20), 4	15 (11–22), 14 ^NS^	16 (11–22), 7 ^NS^	16 (12–19), 7 ^NS^
Haemoglobin A1c, % (range), missing *n*	5.7 (5.4–6.2), 11	5.8 (5.3–6.1), 22 ^**, NS^	5.6 (5.3–5.9), 14 ^NS^	5.7 (5.3–6.0), 13 ^NS^
Troponin T, ng/mL (range), missing *n*	0.019 (0.010–0.033), 26	0.025 (0.010–0.037), 28	0.020 (0.012–0.035), 19	0.023 (0.013–0.037), 22 ^¶¶1^
C-reactive protein, mg/dL (range), missing *n*	0.071 (0.028–0.407), 24	0.126 (0.041–0.377), 18	0.150 (0.040–0.455), 13	0.129 (0.041–0.695), 11 ^¶1, ¶2^
BNP ratio (range), *n* (missing *n*)	4.5 (1.1–15.3), 76 (0)	9.8 (2.7–19.1), 76 (16) **^, NS^	6.4 (1.6–18.0), 76 (7) ^NS^	6.1 (1.4–17.1), 68 (2) ^NS^
pre-BNP ^§^ < 4 (range), *n* (missing *n*)	1.1 (0.8–2.4), 35 (0)	1.8 (1.2–2.9), 35 (14) **^, NS^	1.6 (1.0–2.8), 35 (3) ^NS^	1.5 (0.9–3.1), 33 (0) ^NS^
pre-BNP > 4 (range), *n* (missing *n*)	9.5 (6.3–19.4), 41 (0)	14.8 (9.5–29.5), 41 (2) **^, NS^	16.2 (8.6–32.1), 41 (4) ^NS^	16.0 (9.0–21.0), 35 (2) ^NS^
Death, *n*/patients, *n* (%)		0/76 (0%)	5/76 (7%)	5/68 (7%)
Changes in BNP ratio > 10 compared with baseline (%)		9/76 (12%)	13/76 (17%)	8/68 (12%)

^†^ Period 1: Days 1–125 (4 months) after the first vaccination, ^††^ Period 2: Day 126 to the day of the third vaccination (258 ± 24 days after the first vaccination) in 52 patients or an equivalent period (126–258 days after the first vaccination) in 16 patients who did not receive a third vaccination. ^†††^ Period 3: From the end of Period 2 until 30 June 2022. ^§^ pre-BNP ratio is the mean value of the BNP ratio during the 12 months before the first and second vaccinations. Differences between Period 1 and baseline were analyzed using the Wilcoxon signed-rank test, and differences among Periods 1, 2, and 3 were compared using the Friedman test. ** indicate *p* < 0.01 compared with baseline by the Wilcoxon signed-rank test. NS: No significant difference among the three periods by the Friedman test. ^¶1^ and ^¶2^ indicate *p* < 0.05 compared with Periods 1 and 2, respectively, and ^¶¶1^ indicate *p* < 0.01 compared with Periods 1 by the Friedman test.

**Table 2 jcm-12-01421-t002:** Hazard ratios for the combined outcome.

	**Crude H** **azard Ratio**	** *p* ** **Value**
pre-BNP	1.021 (1.006–1.037)	0.007
Neutralizing antibodies	0.245 (0.083–0.726)	0.011
	**Adjusted** **Hazard Ratio**	** *p* ** **Value**
pre-BNP	1.032 (1.010–1.055)	0.005
Neutralizing antibodies	0.155 (0.041–0.587)	0.006
Age	1.022 (0.957–1.091)	0.51
Male sex	0.597 (0.195–1.824)	0.37
Body mass index	1.069 (0.926–1.234)	0.36
Serum creatinine	0.344 (0.066–1.802)	0.21
Anticoagulation therapy	3.246 (1.106–9.529)	0.032

Time to event was analyzed using a Cox proportional-hazards model using Schoenfeld residuals after fitting. Hazard ratios with 95% confidence interval are described as crude and adjusted hazard ratios for the possible confounders described.

**Table 3 jcm-12-01421-t003:** Relationship between neutralizing antibody levels and various BNP ratios, (**A**) a change in the BNP ratio > 10 in Periods 1–2 compared with baseline, (**B**) a BNP ratio at baseline > 4.0 or (**C**) a BNP ratio > 10 in Periods 1–2.

**A**									
		(BNP ratio in Period 1–2 minus pre-BNP ratio) < 10			(BNP ratio in Period 1–2 minus pre-BNP ratio) > 10				*p* value

Neutralizing antibodies < 220 U/mL		30			14				0.03
Neutralizing antibodies > 220 U/mL		29			3				
**B**					**C**				
		pre-BNP ratio < 4.0	pre-BNP ratio > 4.0	*p* value			BNP ratio in Period 1–2 ^†^ < 10	BNP ratio in Period 1–2 >10	*p* value

Neutralizing antibodies < 220 U/mL		18	26	0.35	Neutralizing antibodies < 220 U/mL		23	21	0.17
Neutralizing antibodies > 220 U/mL		17	15		Neutralizing antibodies > 220 U/mL		22	10	

Fisher’s exact test was used to analyze the relationship of neutralizing antibody levels with the BNP ratio at baseline (pre-BNP ratio), BNP ratio in Periods 1–2, and the difference in BNP ratio between baseline and Periods 1–2. † Brain natriuretic peptide ratio in Periods 1–2 indicates the maximum value of the brain natriuretic peptide ratio in Periods 1 and 2. BNP, brain natriuretic peptide.

## Data Availability

Data can be provided by the corresponding author upon reasonable request.

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
