# Peer review of "Sustained Myocarditis following Messenger RNA Vaccination against Coronavirus Disease 2019: Relation to Neutralizing Antibody and Amelioration by Low-Dose Booster Vaccination"

_jcm, 2023, doi:10.3390/jcm12041421_

Round 1
Reviewer 1 Report
This article is well written.
The legend for the Figure 1 Y-axis plotted data can be improved. What probability is being plotted?
The conclusion states "vaccination-induced cardiac events likely due to autoimmune myocarditis" - this is proposed but not demonstrated. Perhaps changing the wording to "may be due" versus "likely due" would be more appropriate.
This article only has 12 cited references. Additional references related to COVID-19 vaccine myocarditis could be added to the introduction.
Author Response
2 February 2023
Dear Editor and Reviewers,
We thank the reviewers taking the time to review our manuscript (no.jmc-2183523) and providing helpful comments. Please find below our point-by-point responses to the comments.
In addition to responding to the specific comments, we have rearranged the order of the data in Table 3C to match the order in Table 3A and B. Responding to the comments of Reviewer 2, revised manuscript has undergone English proofreading again.
Comment: This article is well written.
Authors’ response: Thank you for this positive comment.
Comment: The legend for the Figure 1 Y-axis plotted data can be improved. What probability is being plotted?
Authors’ response: Thank you for this suggestion. We have revised the legend accordingly.
Comment: The conclusion states "vaccination-induced cardiac events likely due to autoimmune myocarditis" - this is proposed but not demonstrated. Perhaps changing the wording to "may be due" versus "likely due" would be more appropriate.
Authors’ response: We agree with the reviewer and have changed the conclusion as suggested.
Comment: This article only has 12 cited references. Additional references related to COVID-19 vaccine myocarditis could be added to the introduction.
Authors’ response: To address this comment, we have added 5 new references and added further description of COVID-19 myocarditis to the Introduction.
We thank you again for your time and consideration.
Sincerely,
Koji Miyazaki
Reviewer 2 Report
Dear Authors,
This short papered by Miyazaki K et al. investigated the relationships between possible autoimmune myocarditis and doses of booster vaccinations against SARS-CoV2 through the retrospective observational study. Overall, the manuscript has issues in structure, presentation and English. In my opinion, figures and tables are complex, and need more information. The text shows redundancy and abbreviations without introduction. The manuscript including main text needs major modification to convey the information clearly.
The authors already reported a portion of the findings in their previous paper (ref 6) and the audience here could mainly focus on their hypothesis “possible autoimmune myocarditis is reversible through administration of different doses of vaccines”. The patients were separated to three groups based on their BNP ratios. The low-risk group received the normal dose vaccine and other two groups received low dose vaccine. The authors concluded “a low-dose third vaccination ameliorates the risk of cardiac dysfunction compared with normal-dose vaccination, line 224), but the supporting data, especially for handling the controls to lead this conclusion was not clear and need elaboration or redesign of the study.
In line 253, ------ while the changes in the brain natriuretic peptide ratio (Fig. 2) and incidence of cardiac events was decreased compared with after the first and second vaccinations. This description needs to be supported by the statistical analysis and described clearly.
Since there are some publications investigating relationships between myocarditis and COVID-19 vaccine, the authors could discuss and contrast the current and previous findings though the vaccine providers are different.
I believe that the paper provided a few interesting observations and opinions in this field and it would fit to the scope of the journal. However, there are some major issues to be fixed and I will not recommend acceptance of the paper with this current form.
Author Response
2 February 2023
Dear Editor and Reviewers,
We thank the reviewers taking the time to review our manuscript (no.jmc-2183523) and providing helpful comments. Please find below our point-by-point responses to the comments.
In addition to responding to the specific comments, we have rearranged the order of the data in Table 3C to match the order in Table 3A and B. Responding to the comments of Reviewer 2, revised manuscript has undergone English proofreading again.
Reviewer 2
Comment: This short papered by Miyazaki K et al. investigated the relationships between possible autoimmune myocarditis and doses of booster vaccinations against SARS-CoV2 through the retrospective observational study. Overall, the manuscript has issues in structure, presentation and English. In my opinion, figures and tables are complex, and need more information.
Authors’ response: We thank for the reviewer for this comment about the figures and tables. In response, we have simplified the figures and tables by using abbreviations, which are indicated in red in the revised manuscript, and we also added some information, which is highlighted in yellow, to the Figures.
Comment: The text shows redundancy and abbreviations without introduction.
Authors’ response: We checked the text and deleted or revised redundancy (highlighted yellow). We also corrected the abbreviation mRNA and COVID-19 by including the full terms at first appearance in the Introduction.
Comment: The manuscript including main text needs major modification to convey the information clearly. The authors already reported a portion of the findings in their previous paper (ref 6) and the audience here could mainly focus on their hypothesis “possible autoimmune myocarditis is reversible through administration of different doses of vaccines”. The patients were separated to three groups based on their BNP ratios. The low-risk group received the normal dose vaccine and other two groups received low dose vaccine. The authors concluded “a low-dose third vaccination ameliorates the risk of cardiac dysfunction compared with normal-dose vaccination, line 224), but the supporting data, especially for handling the controls to lead this conclusion was not clear and need elaboration or redesign of the study.
Authors’ response: We thank to the reviewer for pointing out the contribution of our study as well as the unclear description to support our conclusion. We agree that handling of the controls is unclear. In the Study protocol (lines 97-102) and Results (line 184-191), it was written as if we evaluated the result of low-dose vaccination in 25 medium-risk patients using the results of normal-dose vaccination in 27 low-risk patients as a sham control. However, we evaluated this using the results of the normal-dose first and second vaccinations of the same patient group as the baseline control. We revised the descriptions in both sections (red and highlighted yellow).
Comment: In line 253, ------ while the changes in the brain natriuretic peptide ratio (Fig. 2) and incidence of cardiac events was decreased compared with after the first and second vaccinations. This description needs to be supported by the statistical analysis and described clearly.
Authors’ response: We agree with the reviewer’s comment. In this comparison, we use the results of 25 medium-risk patients who received the normal-dose first and second vaccinations and low-dose third vaccinations, which means that the patients who died after the normal-dose first and second vaccinations were not included in this comparison, so cardiac events could not be compared. To respond to this comment, we deleted “cardiac events” from this description. We had described only the incidence of worsening of heart failure after the normal-dose first and second vaccinations and low-dose third vaccinations in these 25 medium-risk patients in the Results (lines 207-211). We also deleted it from the Results, because we already described the change in BNP ratio after the normal-dose first and second vaccinations and low-dose third vaccinations in the same patient group (lines 184-188).
Comment: Since there are some publications investigating relationships between myocarditis and COVID-19 vaccine, the authors could discuss and contrast the current and previous findings though the vaccine providers are different.
Authors’ response: We have added some discussion of this and included 5 new references (Nos. 9-13) indicated in red. We have also revised the Introduction to include the same new references.
Comment: I believe that the paper provided a few interesting observations and opinions in this field and it would fit to the scope of the journal.
Round 2
Reviewer 2 Report
The manuscript was largely improved by responding to the questions. Especially, clarification of the controls and additional discussion benefited this improvement. I will recommend acceptance of the paper with this current form.